# The Effect of Epoxy Polymer Addition in Sn-Ag-Cu and Sn-Bi Solder Joints

**DOI:** 10.3390/ma12060960

**Published:** 2019-03-22

**Authors:** Min-Soo Kang, Do-Seok Kim, Young-Eui Shin

**Affiliations:** School of Mechanical Engineering, Chung-Ang University, Seoul 156756, Korea; kang10101@naver.com (M.-S.K.); kdoseok@naver.com (D.-S.K.)

**Keywords:** polymer solder, finite element methods, prediction of life

## Abstract

To analyze the reinforcement effect of adding polymer to solder paste, epoxies were mixed with two currently available Sn-3.0Ag-0.5Cu (wt.% SAC305) and Sn-59Bi (wt.%) solder pastes and specimens prepared by bonding chip resistors to a printed circuit board. The effect of repetitive thermal stress on the solder joints was then analyzed experimentally using thermal shock testing (−40 °C to 125 °C) over 2000 cycles. The viscoplastic stress–strain curves generated in the solder were simulated using finite element analysis, and the hysteresis loop was calculated. The growth and propagation of cracks in the solder were also predicted using strain energy formulas. It was confirmed that the epoxy paste dispersed the stress inside the solder joint by externally supporting the solder fillet, and crack formation was suppressed, improving the lifetime of the solder joint.

## 1. Introduction

Solder joints are exposed to a variety of external environmental conditions, many of which can negatively affect the reliability of these joints via metallurgical and mechanical degradation. It has been found in previous research that the generation of cracks is the main factor affecting the reliability and bonding characteristics of solder joints [1,2,3]. As a result, various studies have been conducted to suppress the appearance of cracks in solder. For example, when solder joints are exposed to high temperatures, hollow Kirkendall voids form due to Cu-Sn diffusion at the solder/Cu interface, leading to the extension of cracks [4,5,6,7]. Research has been conducted with the aim of improving the mechanical properties and suppress the intermetallic compound layer by adding metallic e.g., Ag [8], In [9], Ni [10], Al [11], Co [12] etc. and ceramic e.g., TiO_2_, Al_2_O_3_, ZrO_2_ and Ce_2_O_3_ etc. [11,13,14] particles into the solder or by plating various barrier metals such as Au, Ni, Au/Ni etc., onto the Cu pad. However, these solutions focus only on controlling the suppression of Kirkendall voids while the main determinant of crack formation in solder joints is the difference in the coefficients of thermal expansion (CTEs) for the materials, with cracks appearing at the interface of solder joints. 

Consequently, the difference in CTEs in solder joints needs to be controlled mechanically or metallurgically. In this vein, epoxy solder paste has recently been proposed in order to enhance the bonding characteristics of solder joints. An epoxy solder paste is a homogeneous mixture of plain solder paste and a polymer resin, with the polymer providing a fixation mechanism for the outer region of the solder joint, leading to a higher bonding strength when compared with conventional solder joints. A number of researchers have also studied the effects on solder joint reliability using thermal shock [15], thermal aging [16], and temperature/humidity [17,18] testing to assess the commercial potential lead-free solder materials. However, little is known about the reinforcement mechanisms within epoxy solder joints**.** Therefore, the strengthening effects by epoxy supporting in solder joints were investigated. The solder joints cracks propagation were estimated using finite element analysis (FEA) and experimentation. Also, the FEA results were compared with experimental methods through thermal shock test. 

## 2. Experimental Details

The specimens to be tested were prepared using R3216 chips (Figure 1). R3216 chips offer a general resistance function, consisting of resistive structure coated with a covering layer (Ag) and a termination (Ni barrier). First, the R3216 chips were reflowed onto a printed circuit board (PCB) with either SAC305 (wt.% 96.5Sn3.0Ag0.5Cu) or Sn58Bi (wt.% 42Sn58Bi) solder, which is a low-temperature lead-free solder. The PCB had been finished with an organic solderability preservative. In addition to these control specimens, two types of epoxy solder were used to fabricate the experimental specimens, created by stirring epoxy into either SAC305 or Sn58Bi solder. Epoxy solder must be reflowed at a higher temperature than conventional solder. Melting point of conventional SAC305 solder paste is 221 °C, and that of Sn58Bi solder paste is 139 °C, so the SAC305 epoxy solder was heated to 240 °C for 100 s and the Sn58Bi epoxy solder was maintained at 180 °C for 140 s to facilitate the bonding process. The epoxy was then cured at the outer edge of the solder joint. Each specimen were prepared 18 pieces for test uniformity.

The placement of an engine control unit in the engine and a transmission control unit either on or in the transmission will raise the ambient temperature above 125 °C. To represent automotive conditions, thermal shock tests of the solder/Cu joints were thus performed under C conditions (from −40 °C to 125 °C) based on the JESD22-A106B standard using a shock tester. The target temperatures were maintained for 15 min. 

## 3. Non-Linear Finite Element Analysis (FEA)

FEA modeling was conducted based on R3216 chips. As shown in Figure 2, the model was simplified by modeling 1/4 of the actual chip size (L: 3.2 mm, W: 1.6 mm, H: 0.5 mm), taking advantage of the symmetry of the package. The overall model consisted of a Cu pad, a PCB, a R3216 chip, a Ni electrode, layer, and a solder joint. The R3216 specimen consisted of a 10-μm Ni plating layer on the outer side of a ceramic chip of alumina (Al_2_O_3_), forming an electrode. The thickness of the Cu pad was 30 μm, and the actual measurement value was used. The PCB was modeled by dividing it into five parts to facilitate mesh construction. Solder joints were modeled between the Cu pad and the chip to reflect the final shape of the assembly. In order to conduct FEA, the modeled shape was divided into element types. The thickness of the Ni electrode was modeled as 10 μm. For the thin and plated films, element convergence can be increased by dividing them into node elements with a minimum of five layers. Because this is directly related to the accuracy of the FEA, the solder joints, Ni electrodes, and Cu pads were divided into more than five layers. In addition, components with a large volume, such as the solder joints, were divided into a large number of ellipses to prevent an excessive number of ellipses. The mesh image of the 3D model used in this experiment is presented in Figure 3. Each specimen model was divided into approximately 390,000–410,000 elements. 

The heat generated when using the chip was simulated using FEA. In consideration of this, thermal shock temperatures were applied to the chip as shown in Figure 4. The temperature started and finished at 22 °C (i.e., room temperature), cycling up to 125 °C and down to −40 °C five times, with a 2-minute ramp time and a 15-minute dwell time at each temperature. One cycle thus lasted 30 minutes. 

The boundary conditions of the specimens were limited to 1/4 of the chip scale, using symmetrical constraints, and the same boundary conditions were attached to the sides of the PCB based on the symmetrical structure. The PCB and Cu pad were given fixed support points so that the stresses generated at solder joints could be calculated at the repeated temperatures. To conduct non-linear analysis, the material properties were entered as non-linear values. The basic properties of each material are presented in Table 1.

## 4. Results and Discussion

In order to analyze the reliability of the SAC305 and SAC305 epoxy solder joints, FEA was used to simulate the stress response during thermal cyclic loading. Due to the axial symmetry of the structure, a 1/4 FEA model was employed. Figure 5 shows the stress distribution for conventional solder joints and those formed with the addition of epoxy. During thermal cyclic loading, it was found that maximum stress in the conventional solder joint occurred in the corner of the joint. This is because the difference in the CTE is the largest between the solder and Ni, meaning the thermal stress was concentrated between the solder and the Ni layer. In addition, it can be seen that the greatest stress occurred at the corners of the solder joint and the chip due to the cusp effect. For the specimens with an epoxy fillet, maximum stress occurred at the top of the solder joint, while no maximum stress was generated in the Cu pad/solder/Ni layer. This is likely because the epoxy material, which has a relatively high hardness value, provided support on the outer side, thus dispersing the stress. For this reason, an underfill is often used in ball grid array (BGA) solder joints. Using an underfill encapsulant greatly increases the thermal fatigue life of a solder joint, weakens the effect of stand-off height on reliability, and changes the package deformation mode [19,20,21,22]. 

When thermal loads are applied, the materials undergo thermal expansion. At this time, thermal stress and heat deformation increase at the solder joint. Creep is generally observed when a test specimen is held at high temperatures [23,24,25,26]. However, creep at high temperatures was not observed in this specimen due to the shape and thickness of the solder joint. As the specimen moved into the low-temperature region, the stress-plastic deformation inside the solder decreased. Because the temperature was lowered to −40 °C, the solder joints experienced compressive stress and formed a hysteresis loop (Figure 6). This cyclic hysteresis loop gradually reduced in size each time the cyclic load was applied and can be used as the basis for analyzing fatigue failure. Figure 7 presents the hysteresis loops for the Sn58Bi solder joint; it is clear that they are smaller than the SAC305 solder hysteresis loops. 

Solder fatigue is the mechanical weakening of solder due to deformation under cyclic temperature loading. This can often occur at thermal stress levels below the yield stress of solder as a result of repeated temperature-induced mechanical vibrations, fluctuations, and loads. In fact, the stress in solder joints caused by repetitive thermal loads is lower than the fracture strength of the solder itself. By analyzing the stresses and strains in solder joints, it is possible to predict the lifetime of these joints and the elongation behavior of microcracks. Many studies have been conducted to analyze the thermal creep-fatigue behavior of various solder alloys and develop predictive life damage models using a physics-of-failure approach. These models are used to predict solder joint reliability. 

Solder damage models take a physics-of-failure approach by assessing in terms of cycles to failure a physical parameter that is a critical measure of the damage mechanism process. The relationship between these physical factors and thermal cycles to failure typically takes on a power law or modified power law form with material-dependent model constants. The Darveaux model defines the volume-weighted average inelastic work density, the number of cycles to crack initiation, and the crack propagation velocity over the characteristic cycle leading to the failure of the solder joint. In Equation (1), *N*_0_ represents the number of cycles to crack initiation, Δ*W* represents the inelastic work density, and *K*_1_ and *K*_2_ are material constants. In Equation (2), *da/dN* is the crack corrosion rate, Δ*W* is the inelastic work density, and *K*_3_ and *K*_4_ are the material constants. In this case, the crack propagation speed is considered constant. In Equation (3), *N_f_* represents the number of cycles to failure and *a* represents the characteristic crack length.
(1)$$N_{0}=K_{1}\times \Delta W_{avg}{}^{K_{2}}$$
(2)$$\frac{da}{dN}=K_{3}\times \Delta W_{avg}{}^{K_{4}}$$
(3)Nf=N0+adadN=K1×ΔWavgK2+aK3ΔWavgK4      

The *K* value is an experimentally obtainable value that can be defined based on the shape of the specimen, the shape of the solder joint, the thickness, and the symmetry. The *K* constants for the Darveaux model are shown in Table 2 [27]. Δ*W* can be defined as the area of the hysteresis loop in Figure 6 and Figure 7. The energy accumulated during thermal shock testing is shown in Figure 8. The Δ*W* measured based on the analysis conducted in this paper is shown in Table 3, and the plastic strain density, which is accumulated repeatedly every cycle. 

For the conventional SAC305 solder joints tested in the present study, the plastic deformation energy was calculated to be 81,328 Pa based on stress-plastic deformation, while that for the epoxy resin solder joints was calculated to be 54,292 Pa. Based on the calculated Δ*W*, we used Equation (1) to calculate the number of cycles at which an initial crack would form. As shown in Table 3, cracks would form in conventional SAC305 solder joints after 1032 cycles, compared to 1987 cycles for epoxy SAC305 solder joints. Thus, the initial crack extension time was about twice as long for the epoxy solder. It was predicted that, for the conventional solder joint, the cracks would grow 0.110 μm with every cycle after the initial crack was formed, and the joint would be completely broken after 10,086 cycles. On the other hand, the epoxy solder joint was expected to grow 0.072 μm per cycle, resulting in complete solder joint failure after 15,770 cycles. The epoxy-containing solder was thus 1.5 times more resistant to cyclic thermal loads. 

The plastic deformation energy of Sn58Bi solder joints was calculated to be 75,236 Pa for conventional and 58,417 Pa for epoxy solders. Cracks were calculated to first form in conventional Sn58Bi solder joints after 1171 cycles, growing by 0.102 μm with each subsequent cycle and completely failing after 10,989 cycles. In contrast, the epoxy Sn58Bi solder joint did not begin cracking until after 1764 cycles and reached full fracture after 14,536 cycles. Thus, it was confirmed that crack initiation and growth in the solder joints containing epoxy was slowed, with this resistance maintained even under long-term thermal loads. The strain energy of Sn58Bi solder was found to be lower than SAC305 solder in the conventional solder joints. Thus, resistance to initial cracking is greater for Sn58Bi solder joints. However, there was no significant difference between the two solder types when epoxy was added. 

Cross-sectional images of the fabricated specimens were taken to compare actual crack propagation in thermal shock tests with the life prediction of solder joints calculated using FEA. The cross-sectional images were taken using scanning electron microscopy every 300 cycles during thermal shock testing. All of images of specimens were checked, and the average tendency images were used for analysis. No cracks or other damage were observed in the initial solder joints. During thermal shock testing, cracks were observed in the conventional SAC305 solder joints after 1200 cycles (Figure 9). These cracks expanded into the inner portion of the solder joint as the cycles continued. As shown in Figure 10, cracks in the epoxy SAC305 solder joints were observed after 2000 cycles, which closely followed the FEA results. Generally, the cracks in the conventional solder joints started in the solder under the chip and propagated to the solder fillet or the solder/Ni layer interface. However, if solder joints were bonded with epoxy, the cracks formed in the chip corners. The difference in the crack starting point can be explained by epoxy fillet effects. The epoxy fillet that formed on the outer region of the solder reduced the strain arising from the CTEs and lowered the thermal stress at the finished end of the solder joints. Thus, the cracks started in the inner region of the solder joints. 

The effects of thermal shock testing on conventional Sn58Bi solder joints are shown in Figure 11. Cracks in the upper region of the solder fillet were confirmed after 1200 cycles. These cracks grew into the solder/Ni layer interface as shown in Figure 11e,f. This timing of crack initiation and growth was roughly similar to the FEA of solder joint life. SEM images of the epoxy Sn58Bi solder joints are shown in Figure 12. Voids were observed in the solder joints, but these voids did not affect the solder cracks. Solder cracks were observed after 2000 cycles of thermal shock testing in the upper region of the solder fillet. As shown in Figure 12d, solder cracks were suppressed by the epoxy fillet.

## 5. Conclusions

In this study, repeated thermal stress (cycling between −40 °C and 125 °C) was applied to evaluate the properties of epoxy-containing solder using both experimental and finite element analysis. The specimens consisted of R3216 chips bonded to a PCB using conventional SAC305 (wt.% 96.5Sn-3.0Ag-0.5Cu), conventional Sn58Bi (wt.% 42Sn-58Bi), epoxy SAC305, or epoxy Sn58Bi. The following results were obtained:

(1) The FEA found that thermal stress in the conventional solder joints primarily formed at the bottom of the chip. For the epoxy-containing solder joints, maximum stress was observed at the top of the solder joint, with no maximum stress generated in the Cu pad/solder/Ni layer. The reason for this is that the high mechanical properties of the epoxy material supported the outside of the solder fillet, dispersing the stress in the solder joints.

(2) In the conventional SAC305 solder joints, plastic deformation energy was measured as 81,328 Pa according to the stress-plastic deformation graph, while that of the epoxy solder joint was 54,292 Pa. In the conventional SAC305 solder joints, cracks formed after 1032 cycles, compared to 1987 cycles for their epoxy counterparts. The plastic deformation energy of the conventional Sn58Bi solder joints was 75,236 Pa, while that of epoxy Sn58Bi solder was 58,417 Pa, and crack formation was predicted after 1171 and 1764 cycles, respectively.

(3) In thermal shock testing, it was confirmed that cracks formed in the solder joint. Cracks in the conventional solder joints were observed after 1200 cycles, and cracks in the epoxy solder joints were observed after 2000 cycles. This is in agreement with the number of cycles before crack initiation (*N*_0_) calculated using FEA.

(4) The epoxy in the solder joints was cured on the outside of the solder fillet to ensure the stability of the solder. In addition, at the solder joint interface, where thermal stress due to the difference in the CTEs of the constituent materials occurs, the epoxy fillet reduces deformation and stress, thus increasing the fatigue fracture life inside the solder. As a result, the mechanical reliability of solder joints can be improved.

## Figures and Tables

**Figure 1 materials-12-00960-f001:**
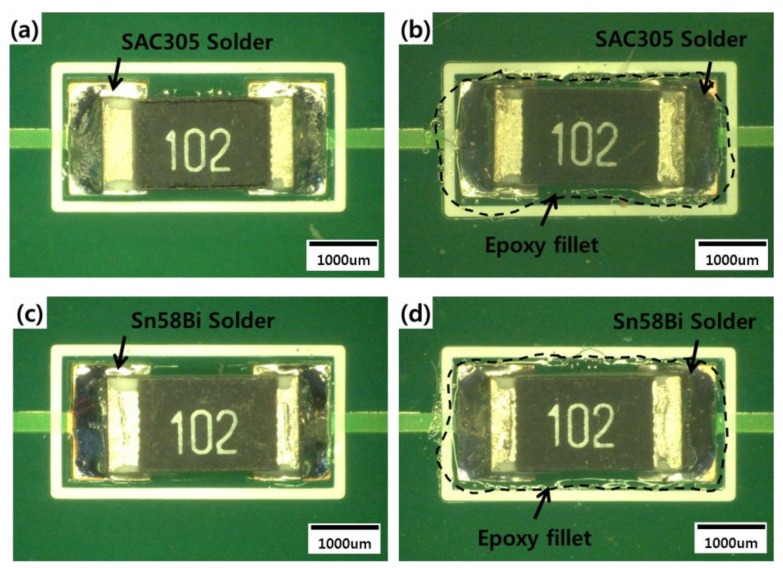
Top view of (**a**) an SAC305 solder joint, (**b**) an SAC305 epoxy solder joint, (**c**) an Sn58Bi solder joint, and (**d**) an Sn58Bi epoxy solder joint.

**Figure 2 materials-12-00960-f002:**
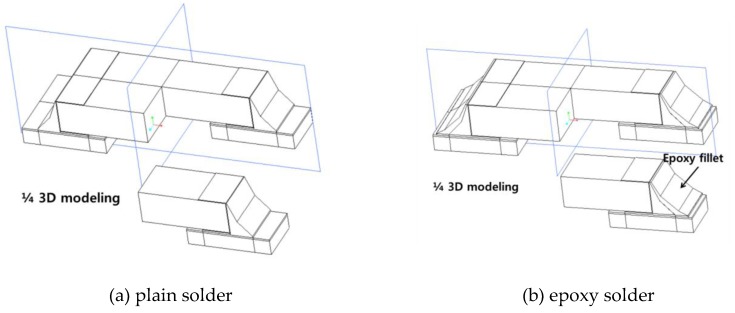
1/4 3D model of plain and epoxy solder specimens. (**a**) plain solder modeling; (**b**) epoxy solder joint modeling.

**Figure 3 materials-12-00960-f003:**
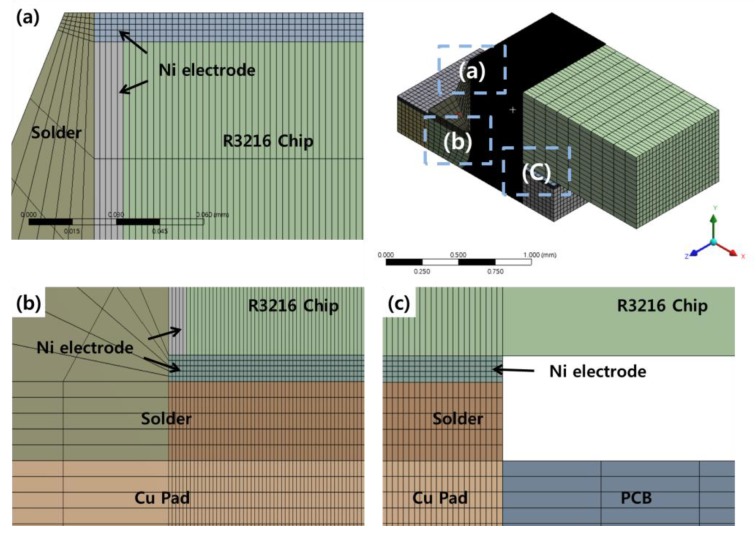
Mesh of a plain solder joint for magnification of region (a), (b), (c). (**a**) Solder joint fillet; (**b**) Ni layer in solder joint (**c**) End of Ni layer in solder joint.

**Figure 4 materials-12-00960-f004:**
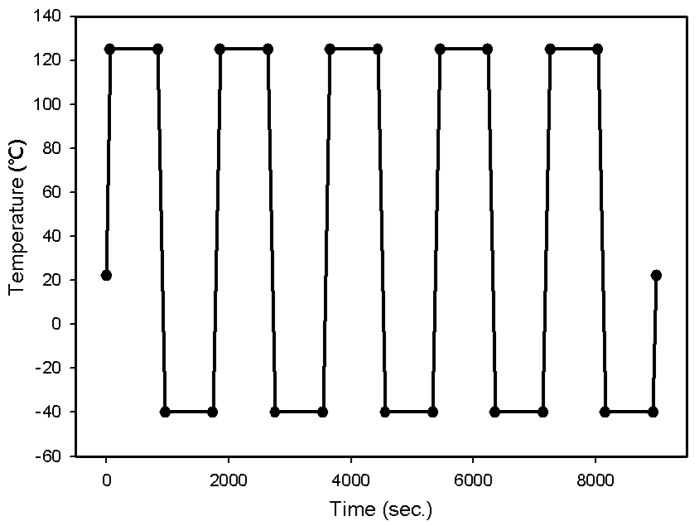
Loading conditions for thermal shock testing.

**Figure 5 materials-12-00960-f005:**
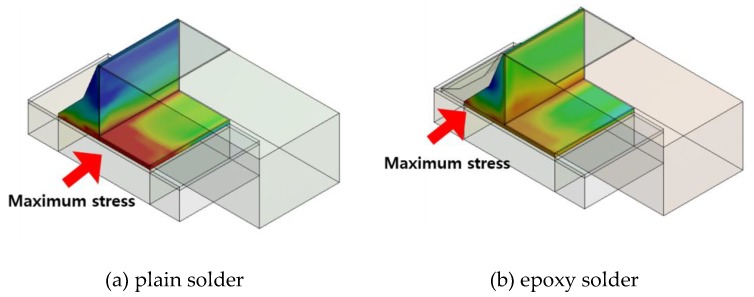
Location of maximum stress in solder joints. (**a**) plain solder; (**b**) epoxy solder.

**Figure 6 materials-12-00960-f006:**
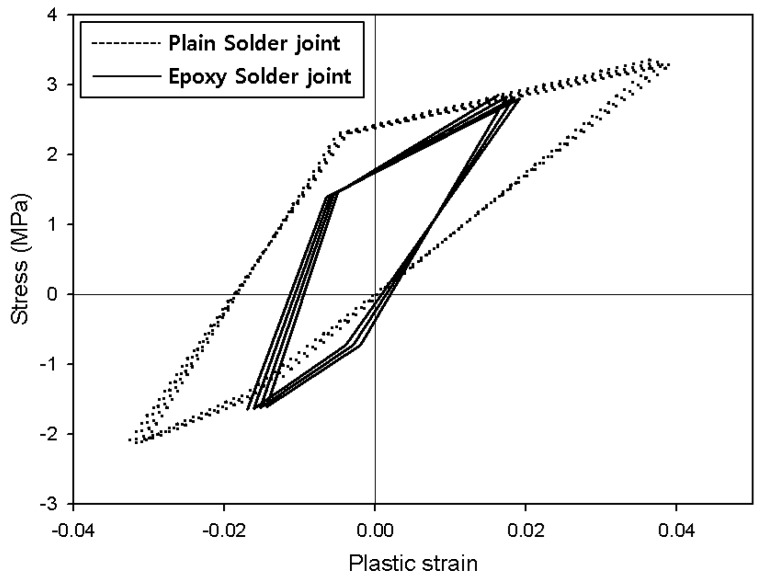
Plastic strain-stress curves for SAC305 solder joints.

**Figure 7 materials-12-00960-f007:**
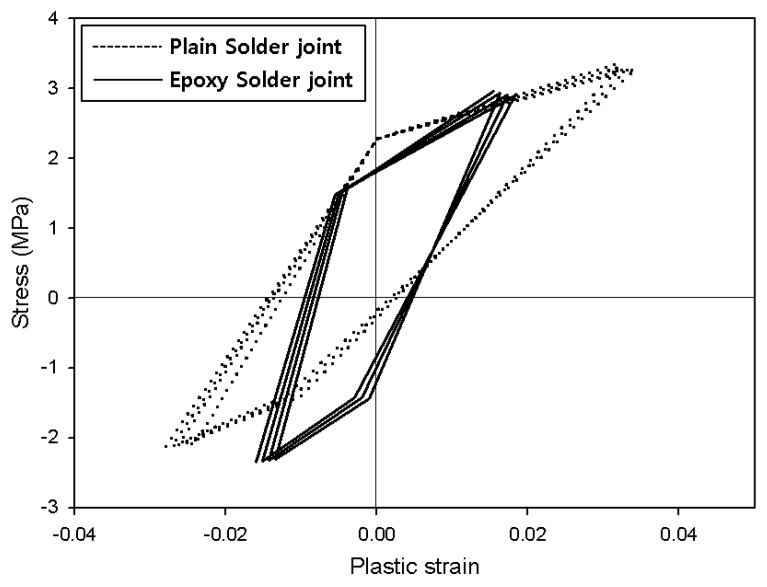
Plastic strain-stress curves for Sn58Bi solder joints.

**Figure 8 materials-12-00960-f008:**
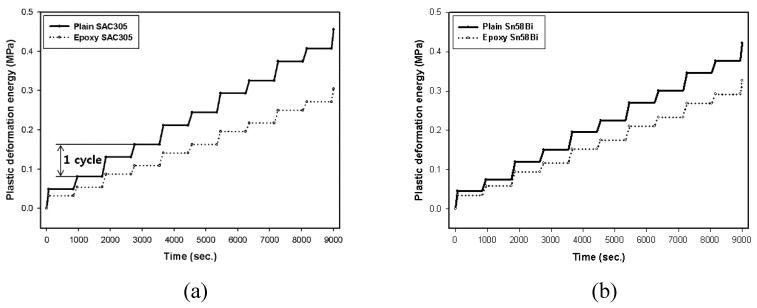
Accumulation of plastic deformation energy during testing: (**a**) SAC305 and (**b**) Sn58Bi solder joints.

**Figure 9 materials-12-00960-f009:**
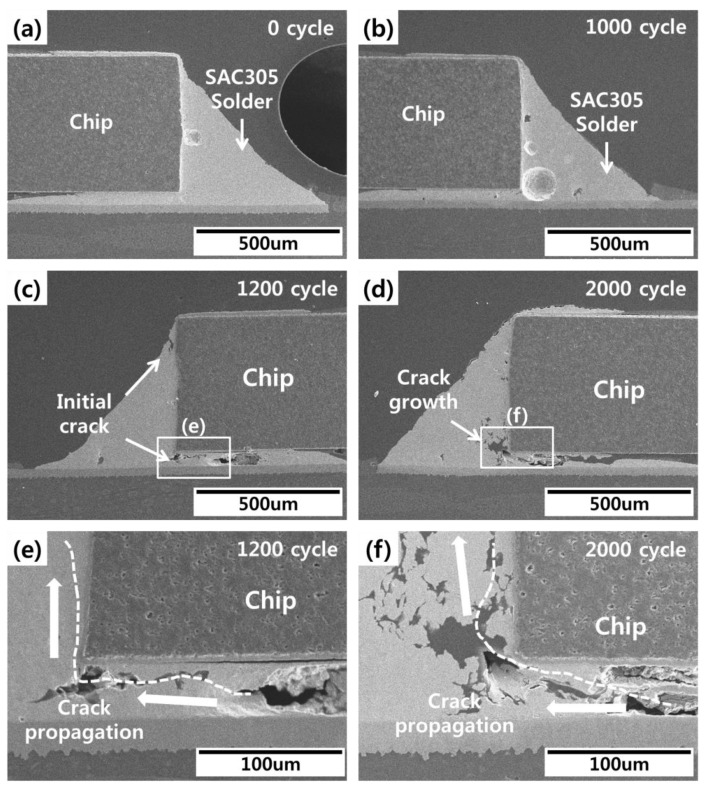
Conventional SAC305 solder joint images during thermal shock testing: (**a**) 0 cycles, (**b**) 1000 cycles, (**c**) 1200 cycles, (**d**) 2000 cycles, (**e**) enlarged image of the initial crack, and (**f**) enlarged image of the crack.

**Figure 10 materials-12-00960-f010:**
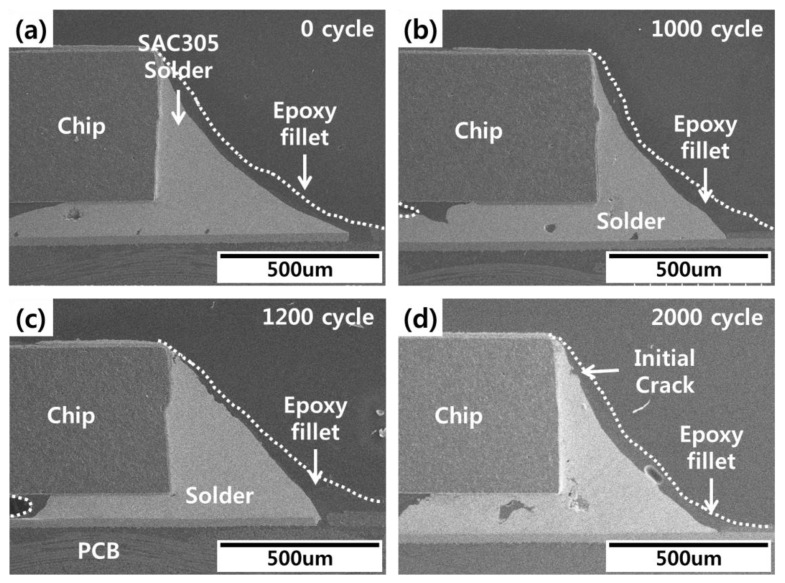
Epoxy SAC305 solder joint images during thermal shock test (**a**) 0 cycles, (**b**) 1000 cycles, (**c**) 1200 cycles, and (**d**) 2000 cycles.

**Figure 11 materials-12-00960-f011:**
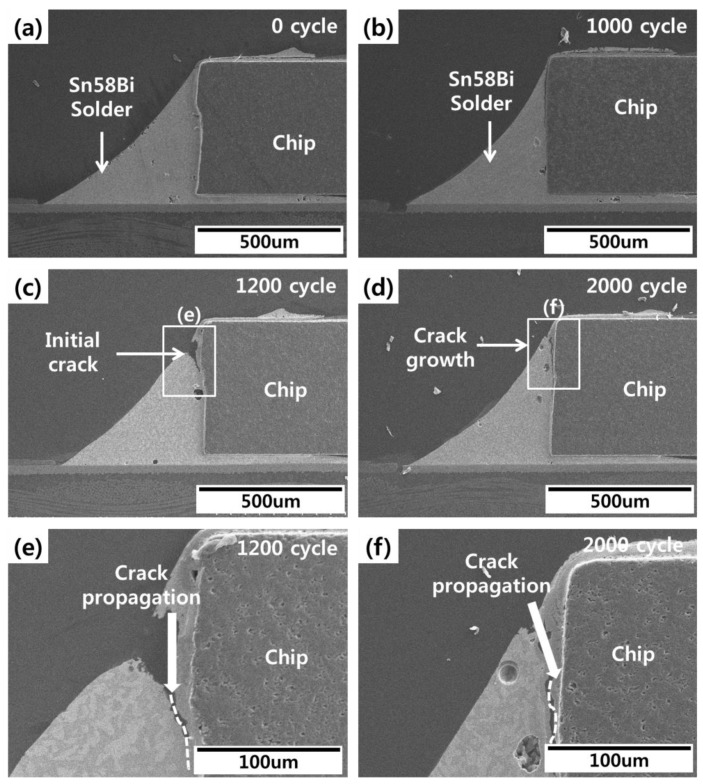
Conventional Sn58Bi solder joint images during thermal shock testing: (**a**) 0 cycles, (**b**) 1000 cycles, (**c**) 1200 cycles, (**d**) 2000 cycles, (**e**) enlarged image of the initial crack, and (**f**) enlarged image of the crack.

**Figure 12 materials-12-00960-f012:**
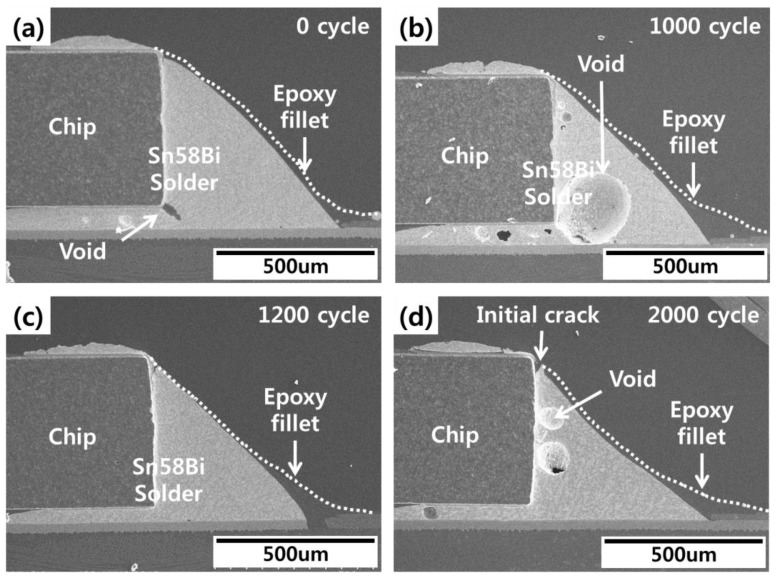
Epoxy Sn58Bi solder joint images during thermal shock testing: (**a**) 0 cycles, (**b**) 1000 cycles, (**c**) 1200 cycles, and (**d**) 2000 cycles.

**Table 1 materials-12-00960-t001:** Properties of the study materials.

Materials	Thermal Conductivity W/(m·K)	Young’s Modulus (GPa)	Poisson’s Ratio	Coefficient of Thermal Expansion CTE μm/(m·K)
Copper	385	76	0.35	16.4
Nickel	60.7	207	0.31	13.1
SAC305	58.7	42	0.35	40.0
Sn58Bi	19	45	0.31	14
PCB	0.81	27	0.39	14
Alumina	30	300	0.22	5.5

**Table 2 materials-12-00960-t002:** K-constants for the Darveaux model.

*K*-Constant	*K*_1_ (cycle/Pa)	*K* _2_	*K*_3_ (mm/cycle∙P)	*K* _4_
Value	9.3 × 10^10^	−1.62	8.64 × 10^−1^^0^	1.04

**Table 3 materials-12-00960-t003:** Lifetime predictions for SAC305 and Sn58Bi solder joints with or without epoxy.

Solder Type	SAC305 Solder		Sn58Bi Solder	
	Conventional	Epoxy	Conventional	Epoxy
**∆W**	81,328 Pa	54,292 Pa	75,236 Pa	58,417 Pa
**N_0_**	1032 cycles	1987 cycles	1171 cycles	1764 cycles
**N_f_**	9054 cycles	13,783 cycles	9,818 cycles	12,773 cycles
**da/dN**	0.110 μm	0.072 μm	0.102 μm	0.78 μm
**Total**	10,086 cycles	15,770 cycles	10,989 cycles	14,536 cycles

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
