# Peer review of "The Effect of Epoxy Polymer Addition in Sn-Ag-Cu and Sn-Bi Solder Joints"

_materials, 2019, doi:10.3390/ma12060960_

Reviewer 1 Report

Dear Editor:          

I would like to express my deep thanks for inviting me to review the manuscript ID: materials-459727

Title:The reinforcement effect of epoxy polymer in lead-free solder joints

Authors: Min-Soo Kang, Do-Seok Kim, Young-Eui Shin

Comments:

Title: The reinforcement effect of epoxy polymer in lead-free solder joints

Replaced by

The effect of epoxy polymer addition in Sn-Ag-Cu and Sn-Bi solder joints

Abstract:

----“with two currently available lead-free solder pastes (SAC305 and Sn58Bi)”

Replaced by

----with two currently available Sn-3.0Ag-0.5Cu (wt.%, SAC305) and Sn-58Bi (wt.%) solder pastes

Introduction part:

Please add references “It has been found in previous research that the generation of cracks is the main factor affecting the reliability and bonding characteristics of solder joints”.

“Research has been conducted with the aim to suppress this process by mixing additives (e.g., Ge, Fe, and S) into the solder [1-4] or by plating various barrier metals onto the Cu pad [5-6]”.

Replaced by

“Research has been conducted with the aim to improve the mechanical properties and suppress the intermetallic compound layer by adding metallic e.g., Ag [1], In [2], Ni [3], Al [4], Co [5] etc. and ceramic e.g., TiO2, Al2O3, ZrO2 and Ce2O3 etc.[4, 6, 7) particles into the solder or by plating various barrier metals such as Au, Ni, Au/Ni etc., onto the Cu pad—“

“A number of researchers have also studied the effects on solder joint reliability using this technique, using thermal shock, thermal aging, thermal cycling, and temperature/humidity testing to”

Replaced by

“A number of researchers have also studied the effects on solder joint reliability and longevity using thermal shock[1], thermal aging [2], and temperature/humidity testing to assess the commercial potential of lead-free solder materials.”

Please add the aim of this work before experimental procedure

Experimental procedure part:

Please mention the purity of SAC and Sn58Bi solder materials.

Results and discussion:

(i) Please combine Figure 9 and Figure 10 and discus the crack propagation behaviour.

(ii) Please combine Figure 11 and Figure 12.

(iii) Please change plastic deformation energy unit to MPa throughout the manuscript

Conclusion part:

Please rewrite the conclusion part concisely.

 RECOMMENDATION

After reviewing the enclosed manuscript for “Materials”, the present manuscript contains some kinds of scientific analysis but it is mandatory required to modify according to the preceding remarks. So, the manuscript can be accepted for publication after minor mandatory revisions have been made.

Author Response

 Title: The reinforcement effect of epoxy polymer in lead-free solder joints

Sentence is modified 

Replaced by

The effect of epoxy polymer addition in Sn-Ag-Cu and Sn-Bi solder joints

----“with two currently available lead-free solder pastes (SAC305 and Sn58Bi)”

Sentence is modified 

Replaced by

----with two currently available Sn-3.0Ag-0.5Cu (wt.%, SAC305) and Sn-58Bi (wt.%) solder pastes

Please add references “It has been found in previous research that the generation of cracks is the main factor affecting the reliability and bonding characteristics of solder joints[1-3]”.

Added the references 

Dongkai S. Analysis of crack growth in solder joints. Soldering and Surface Mount Technology: 1999, 11, 27–32.

John H.; Chang C.; Lee S. Solder joint crack propagation analysis of wafer-level chip scale package on printed circuit board assemblies. IEEE transactions on components and packaging technologies: 2001, 24, 285–292.

Peng Y.; Xiaoyan L.; Xu H.; Liufeng X. Shear strength and fracture mechanism for full Cu-Sn IMCs solder joints with different Cu3Sn proportion and joints with conventional interfacial structure in electronic packaging. Soldering and Surface Mount Technology: 2019, 31, 6–19.

 “Research has been conducted with the aim to suppress this process by mixing additives (e.g., Ge, Fe, and S) into the solder [1-4] or by plating various barrier metals onto the Cu pad [5-6]”.

Sentence is modified 

Research has been conducted with the aim to improve the mechanical properties and suppress the intermetallic compound layer by adding metallic e.g., Ag[8], In[9], Ni[10], Al[11], Co[12] etc. and ceramic e.g., TiO2, Al2O3, ZrO2 and Ce2O3 etc. [11, 13, 14] particles into the solder or by plating various barrier metals such as Au, Ni, Au/Ni etc., onto the Cu pad.

Replaced by

“Research has been conducted with the aim to improve the mechanical properties and suppress the intermetallic compound layer by adding metallic e.g., Ag [1], In [2], Ni [3], Al [4], Co [5] etc. and ceramic e.g., TiO2, Al2O3, ZrO2 and Ce2O3 etc.[4, 6, 7) particles into the solder or by plating various barrier metals such as Au, Ni, Au/Ni etc., onto the Cu pad—“

“A number of researchers have also studied the effects on solder joint reliability using this technique, using thermal shock, thermal aging, thermal cycling, and temperature/humidity testing to”

Sentence is modified 

A number of researchers have also studied the effects on solder joint reliability using  thermal shock[15], thermal aging[16], and temperature/humidity[17,18] testing to assess the commercial potential lead-free solder materials.

Replaced by

“A number of researchers have also studied the effects on solder joint reliability and longevity using thermal shock[1], thermal aging [2], and temperature/humidity testing to assess the commercial potential of lead-free solder materials.”

Please add the aim of this work before experimental procedure

Therefore, in this study, the strength mechanisms and suppression of cracks in epoxy solder joints were investigated using finite element analysis (FEA) and experimentation.

Replaced by

Therefore, the strengthen effects by epoxy supporting in solder joints were investigated. The solder joints cracks propagation were estimated using finite element analysis (FEA) and experimentation. Also, the FEA results were compared with experimental methods through thermal shock test.

The solder composition was explained as SAC305 (wt% 96.5Sn3.0Ag0.5Cu), Sn58Bi (wt% 42Sn58Bi).

(i)  Please combine Figure 9 and Figure 10 and discus the crack propagation behaviour.

(ii)  Please combine Figure 11 and Figure 12.

(iii)  Please change plastic deformation energy unit to MPa throughout the manuscript

The Figure 9, and Figure 10 cannot combine. The Figure 9 is eutectic solder joints, and Figure 10 is Epoxy solder joints. We can see that the solder cracks were declined at epoxy solder joints.

There cannot combine as same reason in the Figure 11 and Figure 12.

If changing the plastic deformation energies unit to MPa, there will be displayed too small. We want to suggest Pa unit.

Reviewer 2 Report

This work showed the study that by adding an epoxy polymer layer to solder pastes, the solder ball fatigue life was improved. In the description of simulation and experiment, there are several details that may need to be reconsidered or confirmed.

Line 86, the ramp rate and dwell time need to be adjusted to be consistent with JEDEC standard or follow the literature's recommendation.

Table 1, the Young's modulus and the CTE should be noticed at certain temperature. And also the plastic property may need to be displayed if possible.

Figure 5 showed the maximum stress, but do not indicate the corresponding temperature and maximum stress value and scale. Without value and scale, the figure was meaningless. It is evident that conventional solder joints has only one boundary between chip and solder, where the maximum stress will happen. But the epoxy one has another layer out of the solder paste. The maximum stress can be happened there.

Line 116, the creep was observed at low temperature is highly suspicious. And the authors explained the phenomenon that was due to the shape and thickness of the solder joint. The explanation was not convincing. Usually the inaccurate material property can lead to this issue.

There is no experiment to validate the prediction of Table 3. Only one or two samples' cross-section image at 1000 or 2000 cycle was not sufficient. Samples have large variations and single result can not demonstrate the simulation prediction. To make your story complete, the Weibull plot with several data points should be included.

Author Response

In the case of thermal shock testing (JESD22-A106B), there is no indication of dwell time.

Using a lifting system, the specimens were moved within 20 seconds from high temperature chamber to low temperature chamber according to the test standard.

The samples were heated or cooled within two minutes, and this time was included in the dwell time.

The position of the maximum stress and the minimum stress varies with the repetitive temperature.

Figure 6, and Figure 7 show the maximum stress at high temperature and the maximum stress at low temperature.

The creep was not only observed at low temperature but also at high temperature.

However, creep at high temperatures was not observed in this specimen due to the shape and thickness of the solder joint.

All test specimens were prepared for eighteen each, and it were mentioned in experimental paragraph.

added sentence

Each specimen was prepared eighteen for test uniformity.

All of images of specimens were checked, the average tendency images were used for analysis.

Round  2

Reviewer 2 Report

The paper was reorganized well with details added to answer review's questions. It was recommended to be published.